# R Bench: Graduate-level Multi-disciplinary Benchmarks for LLM & MLLM Complex Reasoning Evaluation

Meng-Hao Guo [1]   Jiajun Xu [1]   Yi Zhang [1]   Jiaxi Song [1]   Haoyang Peng [1]   Yi-Xuan Deng [1]   Xinzhi Dong [1]
Kiyohiro Nakayama [2]   Zhengyang Geng [3]   Chen Wang [4]   Bolin Ni [5]   Guo-Wei Yang [6]
Yongming Rao [†5]   Houwen Peng [†5]   Han Hu [5]   Gordon Wetzstein [2]   Shi-Min Hu [†✉1]

## Abstract

Reasoning stands as a cornerstone of intelligence, enabling the synthesis of existing knowledge to solve complex problems. Despite remarkable progress, existing reasoning benchmarks often fail to rigorously evaluate the nuanced reasoning capabilities required for complex, real-world problem-solving, particularly in multi-disciplinary and multimodal contexts. In this paper, we introduce a graduate-level, multi-disciplinary, English-Chinese benchmark, dubbed as **R**easoning **Bench** (RBench), for assessing the reasoning capability of both language and multimodal models. RBench spans 1,094 questions across 108 subjects for language model evaluation and 665 questions across 83 subjects for multimodal model testing. These questions are meticulously curated to ensure rigorous difficulty calibration, subject balance, and cross-linguistic alignment, enabling the assessment to be an Olympiad-level multi-disciplinary benchmark. We evaluate many models such as o1, GPT-4o, DeepSeek-R1, *etc*. Experimental results indicate that advanced models perform poorly on complex reasoning, especially multimodal reasoning. Even the top-performing model OpenAI o1 achieves only 53.2% accuracy on our multimodal evaluation. Data and code are made publicly available at here.

## 1. Introduction

*"Setting goals is the first step in turning the invisible into the visible."*      — Tony Robbins

[†]Joint project lead, [1]Tsinghua University, [2]Stanford University, [3]CMU, [4]University of Pennsylvania, [5]Tencent Hunyuan X, [6]Fitten. Correspondence to: Shi-Min Hu <shimin@tsinghua.edu.cn>.

*Proceedings of the 42^{nd} International Conference on Machine Learning*, Vancouver, Canada. PMLR 267, 2025. Copyright 2025 by the author(s).

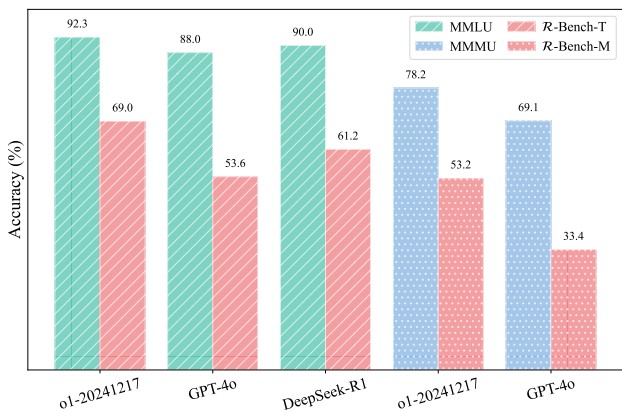

*Figure 1.* Top-1 accuracy comparison of different models on MMLU, MMMU, and RBench. RBench poses a greater challenge to current models.

Reasoning, the systematic process of synthesizing knowledge to solve novel problems, lies at the heart of intelligence. Yet, as foundation models grow increasingly sophisticated, existing benchmarks fail to comprehensively assess their *complex reasoning* capabilities. As shown in the above quote, before equipping foundation models with reasoning skills, we should first define goals for them by establishing a reliable evaluation to assess their reasoning capabilities.

As noted in (Kahneman, 2011) and (Wei et al., 2022), realizing **system-I**, *a.k.a.,* quick and intuitive thinking and **system-II**, *a.k.a.,* slow and deliberate reasoning raises distinct requirements on foundation models. Similarly, assessing quick thinking and complex reasoning requires substantially different assessment methods. On the one hand, evaluating **system-I** needs to evaluate the knowledge and memory, which requires collecting various daily conversations and knowledge-based questions *e.g.,* concept and common sense questions. On the other hand, evaluating **system-II** requires evaluating complex reasoning skills. It requires gathering a diverse range of reasoning questions, such as analytical and deductive ones, which is more chal-

*Table 1.* Comp. denotes comprehensiveness. o1 saturation represents o1 (OpenAI, 2024b) performance on this benchmark. It reflects the challenge that the benchmark poses to advanced models.

| Name | Comp. | o1 Saturation | Language |
|---|---|---|---|
| MMLU | ✓ | 0.923 | en |
| AIME@2024 | ✗ | 0.744 | en |
| $\mathcal{R}$Bench-T | ✓ | 0.690 | en & zh |
| MMMU | ✓ | 0.782 | en |
| $\mathcal{R}$Bench-M | ✓ | 0.532 | en & zh |

lenging to collect and filter than the former. In this paper, we focus on building a reliable complex reasoning benchmark for both large language models (LLMs) and multimodal large language models (MLLMs).

How can we design an ideal assessment for complex reasoning? We believe following four properties are critical.

- **Comprehensiveness.** Evaluating the intelligence of foundation models is akin to evaluating human intelligence. We cannot focus on just one aspect, such as mathematics. A comprehensive evaluation is essential.

- **Difficulty.** A meaningful evaluation should exhibit the capability to effectively discriminate between the performance of different models and provide valuable insights for guiding model improvement. At present, foundation models are developing rapidly, and some simple benchmarks have been saturated and cannot provide guidance and discrimination for advanced models.

- **Multimodality.** We live in a multimodal world, constantly processing various visual and linguistic signals. Therefore, an ideal benchmark should be designed to assess both LLMs and MLLMs.

- **Multilingualism.** We believe that performing complex reasoning is more challenging than understanding multiple languages. A model with robust complex reasoning skills should be capable of solving reasoning problems across different languages. This is like for a human expert, he or she will not lose the ability to address problems due to language changes. Thus, assessing model performance on equally difficult questions across languages is essential. It will provide insight into whether the model has genuinely learned to reason or is merely overfitting to a specific language.

While there have been attempts to create an ideal reasoning benchmark, to the best of our knowledge, existing benchmarks cannot incorporate all four of these key properties simultaneously. Here, we take some widely used reasoning

benchmarks as examples. MMLU (Hendrycks et al., 2021) is a comprehensive benchmark for multi-discipline understanding, which has served as a critical guide for the development of foundation models in recent years. However, considering the current level of model intelligence, this benchmark is close to saturation (*e.g.,* o1 (OpenAI, 2024b) has achieved 92.3% accuracy on it). Besides, it dose not take multimodality and multilingualism into consideration, which is also critical for an ideal reasoning test. MMMU (Yue et al., 2024a) is a holistic evaluation for multimodal reasoning tests. With the launch of o1 (OpenAI, 2024b), this benchmark is also close to saturation. Also, it cannot be used to evaluate language models and ignores multilingual testing. We show the comparison of MMLU, MMMU, and $\mathcal{R}$Bench in Fig. 1 and Tab. 1. Frontiermath (Glazer et al., 2024) collects some challenging problems specifically designed for advanced mathematical reasoning evaluation, which indicates that current models still exhibit weaknesses in mathematical reasoning. However, it falls short in comprehensiveness and multilingual testing. This also applies to Omni-Math (Gao et al., 2024) and AIME (OpenAI, 2024b), both of which serve as benchmarks focused on employing mathematical olympiad challenges.

In this paper, our goal is to build a benchmark $\mathcal{R}$Bench that aligns with the four properties we proposed for evaluating the reasoning abilities of intelligent models. To achieve that, we follow more than 100 college courses from 19 departments at Tsinghua University and collect challenging problems from their exams, textbooks, quizzes, homework, *etc*. After multiple rounds of rigorous screening by experts and models, we finally select 1,094 questions spanning 108 subjects for language models reasoning test, and 665 questions covering 83 subjects for multimodal models reasoning test. We will present the detailed screening process in the Sec 2. After building the $\mathcal{R}$Bench benchmark, we test the reasoning capabilities of various powerful proprietary models such as o1 (OpenAI, 2024b), GPT-4o (OpenAI, 2024a), Gemini (Team et al., 2023), Claude (Anthropic, 2024a), and open-sourced models such as Llama 3 (Touvron et al., 2023), Qwen 2.5 (Yang et al., 2024), *etc*. From experiments, our observations and findings are summarized as follows:

- With the emergence of advanced models like o1, existing multidisciplinary evaluations have nearly reached saturation. Besides, solely relying on math problems, *e.g.,* mathematical olympiad problems, may bring bias in model evaluation. Therefore, the community needs challenging multi-disciplinary benchmarks to guide foundational models in enhancing their reasoning abilities, and the goal of $\mathcal{R}$Bench is to address it.

- We illustrate from three dimensions — expert scoring, model scoring, and model thinking time — that $\mathcal{R}$Bench is a more complex benchmark with higher re-

quirements for model reasoning compared to existing multidisciplinary benchmarks MMLU and MMMU.

- Multimodal complex reasoning remains challenging. Despite rapid advances, models lag behind text-based reasoning. For instance, GPT-4o scores 53.6% on text but only 33.7% in multimodal reasoning on $\mathcal{R}$Bench.

- Chain of Thought (CoT) can enhance reasoning abilities in most chat models, such as GPT-4o. However, for reasoning models like o1-mini, CoT does not have the same effect. This may be because reasoning models inherently build CoT, making explicit CoT ineffective.

- Models maintain high consistency in answering Chinese and English questions of equal difficulty, exceeding 70% for most models, demonstrating strong cross-lingual reasoning capabilities.

- Foundation models perform differently across disciplines. Specifically, GPT-4o achieves 30.4%–68.3% accuracy across various fields.

## 2. $\mathcal{R}$Bench

In this section, we will thoroughly introduce the construction process of $\mathcal{R}$Bench. The entire process involves multiple steps such as data collection, filtering and improving. The overall pipeline is illustrated in Fig. 2.

### 2.1. Data collection

Before gathering reasoning questions, we conducted an investigation of the curriculum systems of graduate and undergraduate students across 19 different departments at Tsinghua University. Based on our survey, we obtained a collection list covering over 100 courses across 19 departments, which is shown in Fig. 2 Step 1.

After acquiring a collection list, we recruit senior undergraduates and graduate students from different departments as experts to provide reasoning question-answer pairs. We recruit a total of 51 experts, with at least two participants from each department, to help us collect and filter questions. During the collection process, we mainly focus on controlling the following key aspects: 1) The questions should align with the collection list we provide. 2) The professional expertise should filter out "knowledge-based" questions—those that rely solely on memory rather than reasoning, such as concept-definition questions. Simultaneously, experts should retain reasoning-based questions and ensure they present a sufficient degree of difficulty. 3) All questions should have corresponding answers that can be automatically verified. In this collecting process, we exclude proof-based questions, as current automated methods cannot verify the correctness of proofs. The process above is shown in Fig. 2 Step 2.

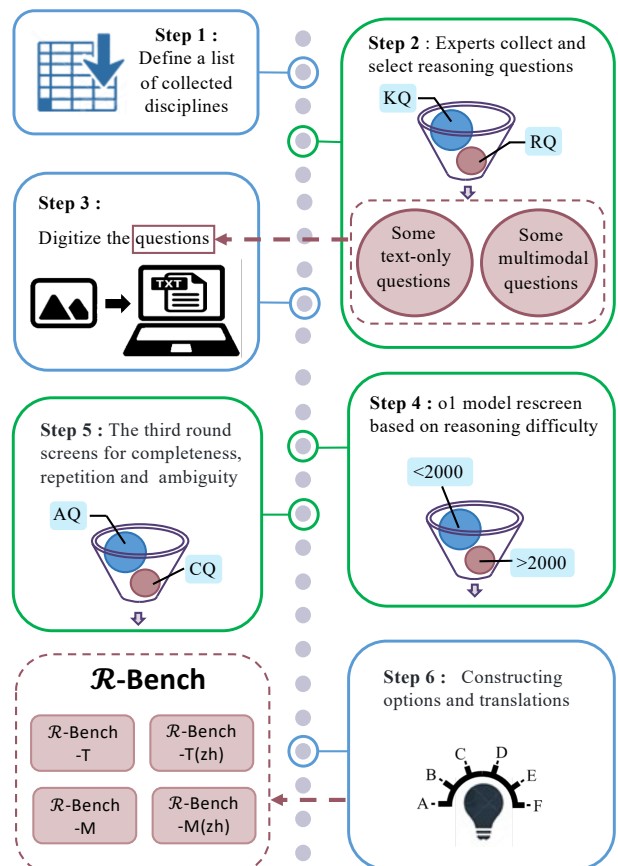

*Figure 2.* Pipeline of building $\mathcal{R}$Bench. The process is divided into six steps, which are detailed in Sec. 2. The funnel represents screening. We always filter out the blue ball and preserve the brown one. In Step 2, KQ and RQ denote knowledge-based questions and reasoning-based questions, respectively. In Step 4, $< 2000$ indicates that the reasoning tokens of o1 are less than 2000. Finally, in Step 5, AQ and CQ represent ambiguous questions and clear questions, respectively. -T indicates text-only testing for LLMs. -M means multimodal testing. zh represents the Chinese version.

After the two steps above, we collected a total of 10,270 questions. Among them, 7,163 questions, which do not include images, are designated for testing language models, while the remaining 3,107 questions, containing images, are allocated for testing multimodal models.

### 2.2. Data digitization

After initially collecting the questions, we find that the collected questions are in a messy format, including pictures, screenshots, text, *etc*. In addition, the summary question files provided by different experts are also different, including pdf, word, excel, *etc*. Therefore, we need to organize and digitize this data.

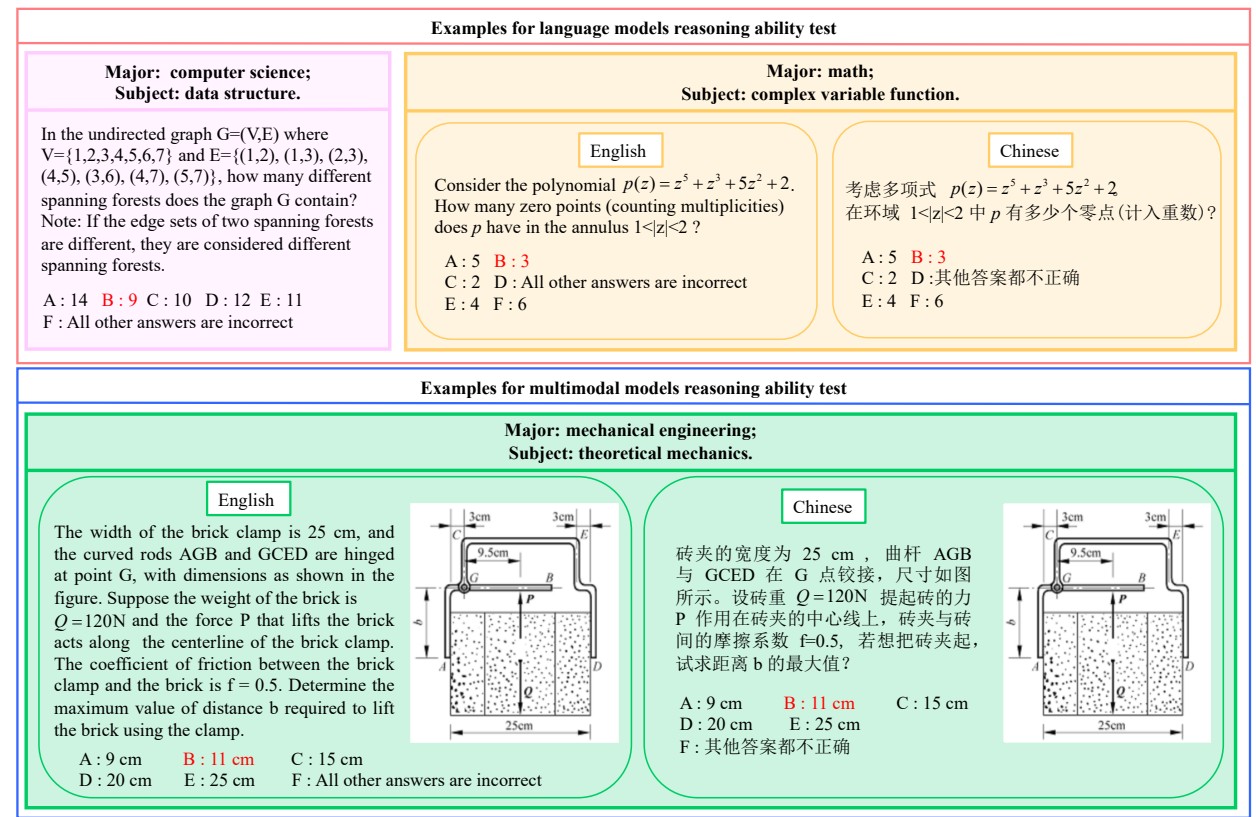

*Figure 3.* Some examples in $\mathcal{R}$Bench. These examples show that $\mathcal{R}$Bench is multidisciplinary, multimodal, and multilingual. As shown in the figure, the problems in $\mathcal{R}$Bench are complex and cannot be solved by quick thinking, which shows that $\mathcal{R}$Bench focuses on deep reasoning problems rather than knowledge problems, such as conceptual problems.

To do so, we recruit a data annotation team of about 20 people. They are responsible for organizing, digitizing, checking, and compiling all the questions into Excel sheets. The questions used for language models are organized in the following format:

"Department - Subject - Question (text) - Answer (text) - Original Question (text, screenshots, photos, *etc.*) - Original Answer (text, screenshots, photos, *etc.*)".

As for questions designed for multimodal models, they are organized into the following format:

"Department - Subject - Question (text) - Answer (text) - Question Images - Original Question (text, screenshots, photos *etc.*) - Original Answer (text, screenshots, photos *etc.*)".

In this process, we utilize tools such as GPT-4o and Mathpix for OCR processing, followed by manual proofreading to ensure it is correct. After the data team organizes the data, we perform a double-check on the OCR results.

## 2.3. Data filtering

As shown in Fig. 2, the funnels in steps 2, 4, and 5 represent three different rounds of data filtering. These three rounds of screening represent expert screening, model-based filtering, and manual review.

**Expert-screening.** As mentioned in Sec. 2.1, we recruit experts from different departments to provide questions for us. They primarily rely on their professional knowledge to filter out "knowledge-based" questions while retaining "reasoning-based" questions.

**Model-screening.** OpenAI o1 (OpenAI, 2024b) is a widely used reasoning model. When we call its API, it returns the number of reasoning tokens, which, to some extent, reflects the difficulty of the question. In this round of screening, we mainly focus on the difficulty of reasoning. We filter out the questions with less than 2,000 reasoning tokens to ensure that our $\mathcal{R}$Bench is a benchmark for reasoning evaluation.

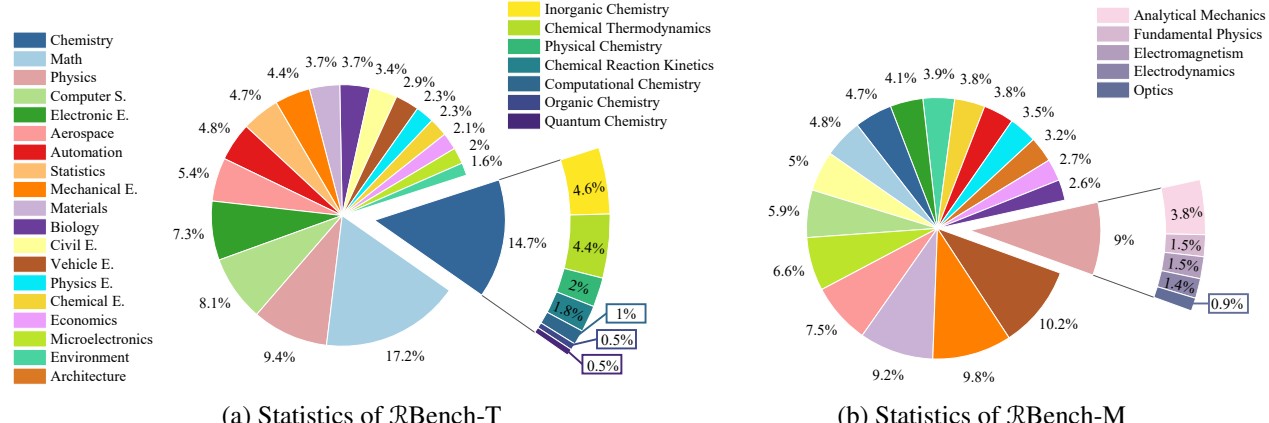

(a) Statistics of $\mathcal{R}$Bench-T     (b) Statistics of $\mathcal{R}$Bench-M

*Figure 4.* According to statistics on $\mathcal{R}$Bench, the benchmark spans 19 departments, including mathematics, physics, biology, computer science, and chemistry, covering over 100 subjects such as Inorganic Chemistry, Chemical Reaction Kinetics, and Electromagnetism. It features 1,094 questions designed for testing language models and 665 questions specifically tailored for evaluating multimodal reasoning capabilities. For a detailed list of subjects, please refer to the appendix.

**Manual review.** Our manual review focuses on whether the question conditions are complete, whether the questions are repeated, whether the questions are ambiguous, and the balance of subjects.

Checking for completeness, repetition, and ambiguity require multiple rounds of thorough review by different individuals to eliminate ambiguities, along with the use of duplication detection tools to avoid repetition. As for the balance check, to reduce testing bias from subject imbalance, we limit the number of questions per subject to a maximum of 50 by filtering out excess.

### 2.4. Conducting options and translations

In order to enable answers to be evaluated automatically and accurately, we convert all questions such as analytical, fill-in-the-blank, and multiple-choice questions into the single-choice question format. We use GPT-4o to construct 5 options for each question and add an option "All other answers are incorrect", which equips each question with 6 candidate answers. Then, we check the options multiple times to ensure the correctness of our construction. Furthermore, we manually adjust the options to ensure a sufficient numerical gap between them, thereby avoiding errors caused by numerical approximations.

Besides, in order to enable $\mathcal{R}$Bench to acquire the multilingual property, we manually constructed English-Chinese translations for each question. During the translation process, we utilize tools like GPT-4o. Each question is meticulously reviewed and refined by three experts fluent in both English and Chinese to ensure correctness and clarity.

### 2.5. Overview of $\mathcal{R}$Bench

After completing the aforementioned steps, we develop $\mathcal{R}$Bench, a graduate-level, multi-discipline, multilingual benchmark designed to evaluate complex reasoning capabilities for both language and multimodal models. Fig. 3 illustrates several examples from $\mathcal{R}$Bench, clearly highlighting its above distinctive features.

$\mathcal{R}$Bench can be divided into four sub-benchmarks: $\mathcal{R}$Bench-T and $\mathcal{R}$Bench-T(zh) for language model evaluation, $\mathcal{R}$Bench-M and $\mathcal{R}$Bench-M(zh) for multimodal model evaluation. Here, $\mathcal{R}$Bench-T denotes $\mathcal{R}$Bench using text-only questions in English for LLM evaluation, whereas $\mathcal{R}$Bench-T(zh) represents $\mathcal{R}$Bench using text-only questions in Chinese for LLM evaluation. Likewise, the other two notations follow the same naming convention.

We conduct statistical analysis on $\mathcal{R}$Benchwith the results presented in Fig. 4. It presents the $\mathcal{R}$Bench-T statistics for text-only questions used in evaluating the reasoning capabilities of language models. $\mathcal{R}$Bench-T spans 18 departments, including mathematics, biology, chemistry, computer science, electronic engineering, and others. It encompasses over 108 subjects, such as calculus, number theory, analytic geometry, ordinary differential equations, and functional analysis, and comprises a total of 1,094 questions. Fig. 4 also presents the statistics of $\mathcal{R}$Bench-M, which evaluates the reasoning capabilities of multimodal models. $\mathcal{R}$Bench-M incorporates a diverse set of question types requiring both textual and visual inputs. It covers 18 departments, such as physics, biology, architecture, and economics, and includes 83 subjects, such as thermodynamics, molecular biology, structural design, and microeconomics, including a total of 665 questions. It is worth noting that we provide English and Chinese versions for all questions.

*Table 2.* Comparison of reasoning requirements for problems in $\mathcal{R}$Bench-T and MMLU via expert and o1 voting.

|  | $\mathcal{R}$Bench-T win | MMLU win | Tie |
|---|---|---|---|
| Expert voting | 85.94% | 10.62% | 3.44% |
| o1 voting | 76.67% | 20.00% | 3.33% |

*Table 4.* The average thinking time of o1 on 30 randomly selected samples from different benchmarks. TT denotes thinking time.

|  | MMLU | $\mathcal{R}$Bench-T | MMMU | $\mathcal{R}$Bench-M |
|---|---|---|---|---|
| TT | 13.5s | 98.2s (7.3×) | 20.3s | 91.7s (4.5×) |

*Table 3.* Comparison of reasoning requirements for problems in $\mathcal{R}$Bench-M and MMMU via expert and o1 voting.

|  | $\mathcal{R}$Bench-M win | MMMU win | Tie |
|---|---|---|---|
| Expert voting | 76.88% | 15.94% | 7.19% |
| o1 voting | 83.33% | 13.33% | 3.33% |

# 3. Experiments

After developing $\mathcal{R}$Bench, we utilize it to assess the complex reasoning capabilities of various LLMs and MLLMs, including both open-source models such as Llama and close-source models such as GPT-4o. Firstly, we aim to demonstrate that $\mathcal{R}$Bench is a benchmark for complex reasoning through expert scoring (user study) and o1 model scoring. Then, we evaluate the reasoning capabilities of models with and without CoT prompting under a zero-shot setting. Finally, we analyze the experiments and summarize observations and findings from the experimental process.

## 3.1. Reasoning comparison with other benchmarks

To illustrate that $\mathcal{R}$Bench is a benchmark designed to evaluate reasoning ability, we employ two methods: expert scoring and reasoning model scoring.

We conducted expert scoring through user studies. To be specific, we randomly selected 30 questions from $\mathcal{R}$Bench-T and another 30 questions from MMLU and presented them to experts for pairwise comparisons to determine which question required more reasoning skills to solve. We constructed similar experiments using the same settings between $\mathcal{R}$Bench-M and MMMU.

For reasoning model scoring, we adopted two approaches. On one hand, we used the o1 model to determine which question required more reasoning ability based on the number of reasoning tokens (reasoning time); on the other hand, we asked the o1 model to directly compare the two questions and determine which one required more reasoning.

The results are shown in Tab. 2,3 and 4. The results indicate that both o1's judgment and the experts' judgment consider $\mathcal{R}$Bench to require significantly higher reasoning ability compared to MMLU and MMMU.

## 3.2. Evaluating reasoning capability of different models

We employ $\mathcal{R}$Bench-T to assess the reasoning capabilities of various LLMs such as o1 (OpenAI, 2024b), GPT-4o (OpenAI, 2024a), DeepSeek-R1 (AI, 2025), Gemini (Team et al., 2024), Claude3.5 (Anthropic, 2024b), Qwen2.5 (Yang et al., 2024), Llama3 (Dubey et al., 2024), etc, in both English and Chinese settings. The evaluation involves utilizing API calls and deploying open-source models locally. For API calls, we utilize the official interfaces with default hyperparameters. For open-source models, we deploy their weights locally using vLLM (Kwon et al., 2023), setting the temperature to 0 while keeping all other parameters at their default values. The evaluation was conducted using the tools provided by OpenCompass (Contributors, 2023). In all tests, the CoT prompt is used by default. For details on the specific prompts, please refer to our appendix. In the results shown in Tab. 5, we found that models designed for reasoning tasks, such as o1, outperform chat models like GPT-4o in complex reasoning. Besides, there remains a significant gap in complex reasoning between open-source models and commercial models.

Moreover, we utilize $\mathcal{R}$Bench-M to evaluate the reasoning capabilities of various MLLMs, including o1 (OpenAI, 2024b), GPT-4o (OpenAI, 2024a), Claude 3.5 (Anthropic, 2024b), Qwen2.5-VL (Yang et al., 2024), and InternVL 2.5 (Chen et al., 2024), etc, across both English and Chinese languages. The evaluation also involves utilizing API calls and deploying open-source models locally. For API calls, we utilize the official interfaces with default hyperparameters. For open-source models, we deploy their weights locally using VLMEvalKit (Duan et al., 2024), setting the temperature to 0 while keeping all other parameters at their default values. In all tests, the CoT prompt is used by default. For details on the specific prompts, please refer to our appendix. We draw three conclusions from the results in Tab. 6. First, we found that models perform worse in multimodal complex reasoning compared to reasoning in a purely linguistic environment. Second, the reasoning model o1 still demonstrates outstanding performance in multimodal complex reasoning evaluation. Third, the gap between open-source and closed-source models is even more pronounced in multimodal complex reasoning.

*Table 5.* Performance comparison of various models on $\mathcal{R}$Bench-T in zero-shot settings with CoT. The table is divided by a middle line: API-based models are listed above the line, while open-source models are shown below. 'zh' indicates the Chinese version.The values in the table represent the Top-1 accuracy, in %.

| Model Name | $\mathcal{R}$Bench-T | $\mathcal{R}$Bench-T(zh) |
| --- | --- | --- |
| o1-20241217 | 69.0 | 70.1 |
| Gemini-2.0-flash-thinking | 68.4 | 67.5 |
| Doubao1.5pro-20250121 | 62.0 | 63.4 |
| o1-preview@20240912 | 62.3 | 62.6 |
| o1-mini@20240912 | 64.0 | 59.9 |
| Doubao-pro-20241215 | 60.7 | 60.8 |
| Claude3.5-sonnet@0620 | 57.5 | 57.0 |
| GPT-4o-20241120 | 53.6 | 51.6 |
| MiniMax-Text-01 | 53.8 | 53.6 |
| GLM-Zero-Preview | 53.6 | 48.6 |
| ERNIE-4.0-8K-Latest | 39.7 | 50.1 |
| Deepseek-R1 | 61.2 | 59.3 |
| Deepseek-V3 | 59.6 | 56.6 |
| Qwen3-235B-A22B | 58.0 | 58.4 |
| Qwen3-32B | 52.3 | 54.3 |
| Qwen2.5-72B-Instruct | 53.7 | 52.0 |
| Llama-3.3-70B-Instruct | 49.5 | 47.6 |
| Qwen2.5-32B-Instruct | 50.8 | 49.9 |
| Gemma-2-27b-it | 36.0 | 38.9 |
| Phi-4-14B | 55.3 | 47.3 |
| Phi-3-14B | 29.5 | 24.4 |
| Qwen3-8B | 47.5 | 45.9 |
| InternLM3-8B-Instruct | 41.1 | 45.8 |
| Qwen2.5-7B-Instruct | 43.6 | 44.5 |
| GLM-4-9b-chat | 25.6 | 32.4 |
| Llama-3.1-8B-Instruct | 26.1 | 23.6 |
| Llama-3.2-3B-Instruct | 24.2 | 24.0 |

*Table 6.* Performance comparison of various models on $\mathcal{R}$Bench-M in zero-shot settings with CoT. The table is divided by a middle line: API-based models are listed above the line, while open-source models are shown below. 'zh' indicates the Chinese version.The values in the table represent the Top-1 accuracy, in %.

| Model Name | $\mathcal{R}$Bench-M | $\mathcal{R}$Bench-M(zh) |
| --- | --- | --- |
| o1-20241217 | 53.2 | 55.0 |
| Claude3.5-sonnet@1022 | 39.7 | 38.3 |
| Doubao1.5pro-20250121 | 37.9 | 42.4 |
| GPT-4o-20241120 | 33.4 | 33.2 |
| Gemini-1.5-Pro | 35.5 | 35.9 |
| Qwen2-VL-72B | 25.1 | 25.7 |
| Qwen2-VL-7B | 19.6 | 22.3 |
| LLaVA-OneVision-7B | 23.8 | 23.5 |
| DeepSeek-VL2 | 21.8 | 24.4 |
| Llama3.2V-11B-Instruct | 20.0 | 18.6 |
| InternVL-2.5-8B | 15.9 | 17.1 |

*Table 7.* Assessing the performance impact of CoT across different models on $\mathcal{R}$Bench-T.

| Model Name | w CoT(%) | w/o CoT(%) |
| --- | --- | --- |
| o1-mini@20240912 | 64.0 | 64.0 |
| GPT-4o-20241120 | 53.6 | 51.5 |
| LLAMA3.3-70B-Instruct | 49.5 | 47.4 |
| Qwen2.5-32B-Instruct | 50.8 | 44.6 |
| Qwen2.5-7B-Instruct | 43.6 | 42.6 |

focused models such as o1-mini. We conjecture that this discrepancy arises because reasoning models inherently utilize CoT-like mechanisms, leading to the explicit addition of CoT redundant and ineffective.

### 3.3. Observations and findings

**Multimodal reasoning remains challenging for current models.** We compared the performance of the same model on $\mathcal{R}$Bench-T and $\mathcal{R}$Bench-M. For example, o1 achieved 69.0% on $\mathcal{R}$Bench-T but only 53.2% on $\mathcal{R}$Bench-M. The same situation also occurs in other models, such as GPT-4o. It indicates that the model's capability in language reasoning significantly surpasses its ability in multimodal reasoning. Therefore, a key focus of research in the recent future will be how to transfer linguistic intelligence to the multimodal domain.

**The effect of CoT.** In Tab. 7, we tested the effect of CoT on five models. As seen in the table, most models benefit from CoT, but it has no impact on o1-mini. The results indicate that CoT enhances the performance of chat models like GPT-4o. However, it has no notable impact on reasoning-

**Consistency between English and Chinese questions.** As shown in Fig. 5, we tested the consistency of the same question across different languages on $\mathcal{R}$Bench-T. It can be observed that most models, such as o1, Doubao1.5pro, and GPT-4o, exhibit a certain degree of consistency across different languages. This suggests that foundation models have already demonstrated a certain level of intelligence, enabling them to perform reasoning on problems of the same difficulty in different linguistic environments. However, these models are not perfect and still require further improvement in this aspect. This consistency reflects the extent to which the model overfits different languages. Therefore, we hope future models will focus more on learning how to reason rather than merely fitting to specific languages.

**Models show significant performance variation across disciplines.** Fig. 6 shows the performance of GPT-4o in

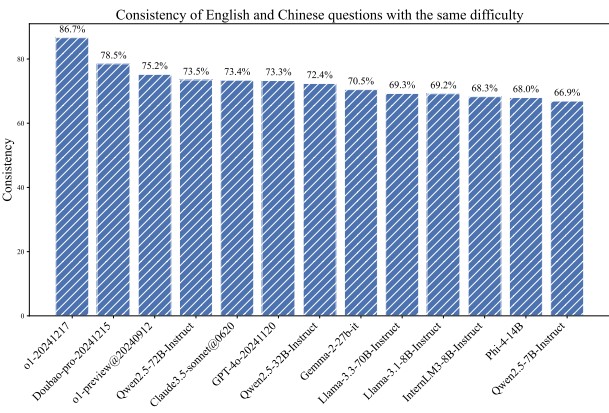

*Figure 5.* The performance of different models on questions of the same difficulty in Chinese and English.

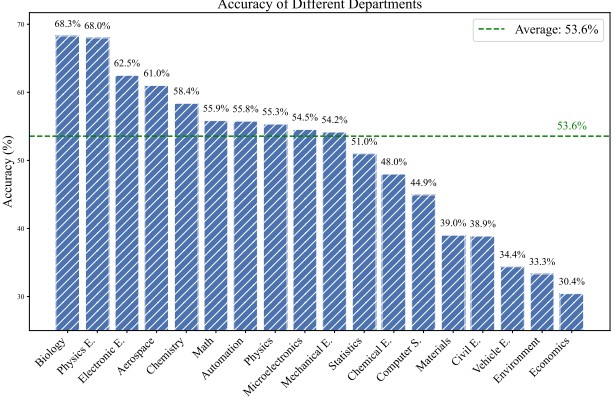

*Figure 6.* GPT-4o on ℛBench-T across different departments, which shows large variation among different disciplines..

different areas of the ℛBench-T benchmark. From the figure, it can be observed that the performance varies significantly across different domains, with a range reaching 37.9%. This suggests that if we want to improve the reasoning ability of models, we should take a comprehensive approach rather than focusing solely on improvements in a single subject, such as mathematics.

## 4. Related Work

### 4.1. Foundation models

With the emergence of ChatGPT (OpenAI, 2022), foundation models (Ouyang et al., 2022; Touvron et al., 2023; Yang et al., 2024; Jiang et al., 2023; Team et al., 2024; Zeng et al., 2022; Bi et al., 2024; Liu et al., 2024a; Cai et al., 2024; Anthropic, 2024a; Wu et al., 2024) are increasingly being leveraged across diverse fields such as writing, coding, education, healthcare , finance, and more, serving as a source for providing intelligence. Now, foundation models have become an essential part of our daily work and life.

To build a high-quality foundation model, we believe that five key aspects are essential: pre-training (Vaswani et al., 2017; Raffel et al., 2020; Sun et al., 2023; Radford et al., 2021; 2018), supervised fine-tuning (Ouyang et al., 2022; Liu et al., 2024b; Pareja et al., 2024; Li et al., 2024; Zhu et al., 2023; Taori et al., 2023), preference optimization (Rafailov et al., 2024; Schulman et al., 2017; Lightman et al., 2023; Pal et al., 2024; Azar et al., 2024),test-time enhancement (Brown et al., 2020; Wei et al., 2022; OpenAI, 2024b; DeepSeek, 2024; Dong et al., 2022), and trustworthy evaluation (Chiang et al., 2024; Li et al., 2023; Chen et al., 2021; Jain et al., 2024; Yu et al., 2023). Reliable evaluation plays a crucial role in revealing models' weaknesses and shortcomings, guiding further optimization and improvement, which is also the focus of this paper.

### 4.2. Evaluation for foundation models

Evaluating the intelligence of foundation models is a multifaceted and complex challenge, akin to assessing human intelligence. Researchers have introduced various evaluation benchmarks, broadly categorized into fast-thinking assessments ( *a.k.a.,* system-I evaluation (Chiang et al., 2024; Dubois et al., 2024; Zheng et al., 2023; Lin et al., 2021; 2024; Lu et al., 2022; Yu et al., 2023; Li et al., 2023), which requires the foundation models to memorize extensive knowledge and retrieve efficiently, and slow-thinking assessments (*a.k.a.,* system-II evaluation (Hendrycks et al., 2021; Yue et al., 2024a; Lu et al., 2023; Wang et al., 2024a; Liu et al., 2024c; Chen et al., 2021; Jimenez et al., 2023), which emphasizes the complex reasoning skills of foundation models. ℛBench focuses on the latter.

MMLU (Hendrycks et al., 2021) is the pioneer in reasoning evaluation, which proposes a multi-discipline understanding test. After that, lots of multi-discipline benchmarks (Rein et al., 2023; Wang et al., 2024b; Yue et al., 2024b) at the undergraduate or graduate level are proposed for reasoning assessment. However, with the rapid development of intelligent models (OpenAI, 2024b), these benchmarks are close to saturation. Besides, several studies (Glazer et al., 2024; Gao et al., 2024; Lu et al., 2023; Wang et al., 2024a) assess reasoning ability through complex mathematical problems such as mathematical olympiad challenges, which bring challenges and guidance to current foundation models. However, only guiding the model to improve its mathematical reasoning skills appears to be limited. In this paper, our target is to build a reliable benchmark for LLM and MLLM reasoning evaluation, which matches the comprehensiveness of MMLU (Hendrycks et al., 2021) while achieving

the difficulty of mathematical olympiad questions.

# 5. Conclusion

In this paper, we proposed $\mathcal{R}$Bench, a graduate-level multi-disciplinary, multilingual benchmark for both LLM and MLLM reasoning evaluation, which has coverage similar to MMLU and MMMU while reaching the difficulty of mathematical competitions such as AIME@2024. We evaluated multiple closed-source and open-source models such as OpenAI o1, GPT-4o, DeepSeek-R1, etc, on $\mathcal{R}$Bench and observed both the progress and limitations of current models in reasoning. Later, we will make the data and code available, hoping to provide guidance and insight for the development of foundation models.

# Acknowledgement

This work was supported by the National Natural Science Foundation of China (project No. 62495061, 623B2057), the Research Grant of Tsinghua-Tencent Joint Laboratory for Internet Innovation Technology.

# Impact Statement

$\mathcal{R}$Bench provides actionable insights into model reasoning gaps, especially in multimodal and cross-lingual contexts, and helps guide the development of more generalizable and robust reasoning models. Moreover, our benchmark enables researchers and practitioners to diagnose and compare reasoning abilities across a broad range of disciplines and modalities, supporting not only academic progress but also downstream applications in education, healthcare, law, and science. All data and code are publicly released to foster transparency and reproducibility, making R-Bench a valuable asset for the community.

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

# A. Appendix.

## A.1. Response Example

Here, we present our question and the output of the OpenAI's o1 model in the box below. The model carried out a step-by-step and detailed reasoning based on the question and finally arrived at the correct answer.

---

**Question and the response by o1**

Answer the following single choice question. The last line of your response should be of the following format: 'ANSWER: $LETTER' (without quotes) where LETTER is one of Options(e.g. one of ABCDEF). Think step by step before answering. Question: A sudden environmental change causes a recessive allele to become completely lethal. How many generations will it take for its frequency to decrease from $10^{-2}$ to $10^{-3}$?
A. 950
B. 850
C. All other answers are incorrect
D. 900
E. 800
F. 920
Answer:

### Response by o1
To solve this, use the fact that for a fully recessive lethal allele (with initial frequency $q_0$), its frequency in the next generation is given by:

$$q_{n+1} = \frac{q_n}{1 + q_n}$$

Define $R_n = \frac{1}{q_n}$. Then from the recurrence above:

$$
\begin{aligned}
R_{n+1} &= \frac{1}{q_{n+1}} \\
&= \frac{1 + q_n}{q_n} \\
&= \frac{1}{q_n} + 1 \\
&= R_n + 1
\end{aligned}
$$

Thus, $R_n = R_0 + n$. Since $R_0 = \frac{1}{q_0}$ and we want $q_n$ to decrease from 0.01 to 0.001:

- $R_0 = \frac{1}{0.01} = 100$.

- We need $\frac{1}{q_n} = 1000$, so $R_n = 1000$.

- Hence $1000 = 100 + n$, giving $n = 900$.

ANSWER: D

---

## A.2. CoT Prompt

Here, we present the format of our CoT prompt in following box. As we can see, our CoT prompt mainly uses "Think step by step", which shows that even a simple prompt still

has a positive effect on most chat models.

---

**System Prompt for ℛBench**

Answer the following single-choice question. The last line of your response should be of the following format: 'ANSWER: $LETTER' (without quotes) where LETTER is one of the Options (e.g. one of ABCDEF). Think step by step before answering.
Question: {Question Input}
A. {OptionA}
B. {OptionB}
C. {OptionC}
D. {OptionD}
E. {OptionE}
F. {OptionF}
Answer:

### Example:
Answer the following single choice question. The last line of your response should be of the following format: 'ANSWER: $LETTER' (without quotes) where LETTER is one of Options(e.g. one of ABCDEF). Think step by step before answering.
Question: Consider a CMOS inverter driving a wire of length L. In the initial design, the on-resistance of the inverter is equal to the total resistance of the wire, the source-drain capacitance of the inverter is equal to the total capacitance of the wire, and the total delay of the inverter and wire is tp. Now, the devices are scaled down using Constant Field Scaling, while the wire is ideally scaled down. Assuming the wire can be modeled using a lumped parameter model, answer the following questions in a first-order approximation:
(1) Assuming the wire is a local wire, and the scaling factors for both process and supply voltage are 2, express the total delay after scaling in terms of tp.
(2) Now assume the wire is global, and the length of the wire increases inversely with the process scaling, with scaling factors for both process and supply voltage being 2, express the total delay after scaling in terms of tp.
A. (1) 1/3tp (2) 35/6tp
B. (1) 1/2tp (2) 38/6tp
C. All other answers are incorrect
D. (1) 3/4tp (2) 36/6tp
E. (1) 5/6tp (2) 39/6tp
F. (1) 2/3tp (2) 37/6tp
Answer:

---

### A.3. Specific subject distribution

We present the specific subject distributions of ℛBench-T and ℛBench-M in Table 8 and Table 8, respectively. It can be observed that ℛBench has a broad coverage, making it difficult to improve performance on ℛBench by overfitting to specific subjects.

Table 8: Distribution of Courses by Discipline in $\mathcal{R}$Bench-T

| Discipline | Specific Subject | Count |
|---|---|---|
| Civil Engineering | Fluid Mechanics | 33 |
| | Structural Mechanics | 2 |
| | Surveying | 1 |
| Chemical Engineering | Principles of Process Transport | 3 |
| | Principles of Chemical Engineering | 15 |
| | Analytical Chemistry | 7 |
| Economics | Advanced Mathematical Economics | 1 |
| | Econometrics | 3 |
| | Intermediate Financial Theory | 1 |
| | Financial Engineering | 1 |
| | Game Theory and Mechanism Design | 5 |
| | Intermediate Microeconomics | 2 |
| | Principles of Accounting | 2 |
| | Time Series Analysis | 8 |
| Biology | Soil Science | 1 |
| | Genetics | 16 |
| | Physiology | 1 |
| | Biochemistry | 15 |
| | Heredity | 8 |
| Physics | Mathematical Methods in Physics | 10 |
| | Electromagnetics | 11 |
| | Optics | 23 |
| | Quantum Mechanics | 8 |
| | Analytical Mechanics | 6 |
| | Electrodynamics | 5 |
| | Thermodynamics and Statistical Physics | 6 |
| | General Relativity | 2 |
| | Basic Physics | 29 |
| | Group Theory | 3 |
| Aerospace | Mechanics of Materials | 1 |
| | Structural Mechanics of Aircraft | 3 |
| | Theoretical Mechanics | 2 |
| | Fluid Mechanics and Aerodynamics | 15 |
| | Optimal Control | 29 |
| | Propulsion Principles and Thermal Fluid Basics | 9 |
| Microelectronics | Digital Large - Scale Integrated Circuits | 20 |
| | Analog Circuits | 2 |
| Automation | Signals and Systems | 20 |
| | Operations Research | 15 |
| | Automatic Control Theory | 17 |
| Electronic Engineering | Communication and Network | 15 |
| | Principles of Analog Circuits | 4 |
| | Electromagnetic Fields and Waves | 8 |
| | Fundamentals of Solid State Physics | 12 |
| | Digital Signal Processing | 8 |
| | Stochastic Processes | 3 |

Table 8 – *Continued from previous page*

| Discipline | Specific Subject | Count |
|---|---|---|
| | Solid State Physics | 4 |
| | Applied Stochastic Processes | 26 |
| Mechanical Engineering | Fluid Mechanics | 11 |
| | Principles and Interface Technology of Single - Chip Microcomputer | 4 |
| | Mechanical Design | 1 |
| | Theory of Machines | 4 |
| | Electromechanical Transmission and Control | 4 |
| | Electrical and electronic Technology | 2 |
| | Mechanical Vibration | 3 |
| | Hydraulic and Pneumatic Transmission | 1 |
| | Engineering Thermodynamics | 8 |
| | Mechanics of Materials | 9 |
| | Theoretical Mechanics | 1 |
| Computer Science | Data Structure | 24 |
| | Combinatorial Mathematics | 3 |
| | Numerical Analysis | 5 |
| | Cryptography | 17 |
| | Automata | 8 |
| | Principles of Computer Organization | 2 |
| | Compilation Principles | 5 |
| | Computer Network | 11 |
| | Operating System | 11 |
| | Computer Architecture | 3 |
| Chemistry | Inorganic Chemistry | 50 |
| | Chemical Thermodynamics | 48 |
| | Chemical Reaction Kinetics | 20 |
| | Introduction to Computational Chemistry | 11 |
| | Quantum Chemistry | 5 |
| | Physical Chemistry | 22 |
| | Organic Chemistry | 5 |
| Mathematics | Complex Analysis | 26 |
| | Analytic Geometry | 26 |
| | Advanced Calculus | 9 |
| | Number Theory | 22 |
| | Matrix Analysis | 36 |
| | Partial Differential Equations | 29 |
| | Mathematical Analysis | 9 |
| | Stochastic Differential Equations | 8 |
| | Functional Analysis | 14 |
| | Ordinary Differential Equations | 3 |
| | Differential Geometry | 3 |
| | Topology | 3 |
| Physics Engineering | Thermodynamics and Statistical Physics | 5 |
| | Nuclear Radiation Physics and Detection | 20 |
| Materials | Quantum and Statistics | 14 |
| | Physical Properties of Materials | 1 |
| | Fundamentals of Materials Science | 23 |

Table 8 – *Continued from previous page*

| Discipline | Specific Subject | Count |
|---|---|---|
| | Materials Analysis and Characterization | 3 |
| Vehicle Engineering | Finite Element Analysis Basics | 1 |
| | Principles of Automotive Power System | 11 |
| | Automotive Electronics and Control | 1 |
| | Fundamentals of Control Engineering | 3 |
| | Theory of Automobile | 3 |
| | Automobile Construction | 1 |
| | Discrete Mathematics | 12 |
| Statistics | Probability Theory | 42 |
| | Introduction to Bayesian Statistics | 6 |
| | Reliability Data and Survival Analysis | 1 |
| | Statistical Inference | 2 |
| Environment | Principles of Environmental Engineering Science and Engineering | 5 |
| | Water Treatment Engineering | 12 |
| | Environmental Chemistry | 1 |

Table 9: Distribution of Courses by Discipline in $\mathcal{R}$Bench-M

| Discipline | Specific Subject | Count |
|---|---|---|
| Aerospace | Materials Mechanics | 26 |
| | Fluid Mechanics and Aerodynamics | 2 |
| | Theoretical Mechanics | 16 |
| | Aircraft Structural Mechanics | 2 |
| | Propulsion Principles and Thermal Fluids | 2 |
| | Optimal Control | 2 |
| Integrated Circuits | Digital VLSI | 11 |
| | Analog Circuits | 19 |
| | Digital Electronics Fundamentals | 8 |
| | Analog Electronics Fundamentals | 6 |
| Chemical Engineering | Transport Process Principles | 2 |
| | Chemical Principles | 11 |
| | Physical Chemistry | 12 |
| Chemistry | Chemical Thermodynamics | 1 |
| | Chemical Reaction Kinetics | 10 |
| | Inorganic Chemistry | 1 |
| | Organic Chemistry | 17 |
| | Physical Chemistry | 2 |
| Computer Science | Data Structures | 4 |
| | Combinatorics | 1 |
| | Discrete Mathematics | 12 |
| | Theory of Automata | 7 |
| | Operating Systems | 6 |
| | Compilers | 4 |
| | Computer Architecture | 3 |
| | Cryptography | 1 |
| | Computer Networks | 1 |
| Physics | Analytical Mechanics | 25 |
| | Optics | 6 |
| | Electrodynamics | 9 |
| | Electromagnetism | 10 |
| | Basic Physics | 10 |
| Electrical Engineering | Analog Circuit Principles | 17 |
| | Signals and Systems | 3 |
| | Digital Signal Processing | 1 |
| | Communication and Networks | 4 |
| | Electromagnetic Fields and Waves | 1 |
| | Solid State Physics | 1 |
| Mathematics | Complex Analysis | 8 |
| | Analytic Geometry | 2 |
| | Probability Theory | 5 |
| | Stochastic Processes | 3 |
| | Analysis | 1 |
| | Probability and Statistics | 1 |
| | Mathematical Analysis | 5 |
| | Statistics | 6 |

Table 9 – *Continued from previous page*

| Discipline | Specific Subject | Count |
|---|---|---|
| | Topology | 1 |
| Environmental Engineering | Water Treatment Engineering | 4 |
| | Environmental Science and Engineering Principles | 9 |
| | Environmental Monitoring | 6 |
| | Water Pollution Control Project | 4 |
| | Environmental Chemistry | 1 |
| | Solid Waste Treatment and Disposal | 2 |
| Biology | Genetics | 13 |
| | Biochemistry | 4 |
| Material Science | Materials Mechanics | 37 |
| | Quantum and Statistical Mechanics | 5 |
| | Material Analysis and Characterization | 1 |
| | Basic Material Science | 18 |
| Economics | Operations Research | 2 |
| | Principles of Accounting | 3 |
| | Financial Engineering | 2 |
| | Intermediate Financial Theories | 3 |
| | Game Theory and Mechanism Design | 8 |
| Mechanical Engineering | Electrical and Electronics Technology | 5 |
| | Mechanical Design | 24 |
| | Electromechanical Transmission and Control | 1 |
| | Hydraulic and Pneumatic Transmission | 9 |
| | Mechanical Vibrations | 7 |
| | Fluid Mechanics | 7 |
| | Principles of Mechanics | 1 |
| | Theoretical Mechanics | 11 |
| Civil Engineering | Structural Mechanics | 33 |
| Engineering Physics | Engineering Mechanics | 23 |
| Automation | Automatic Control Theory | 25 |
| Vehicle Engineering | Fluid Mechanics | 34 |
| | Engineering Control Basics | 8 |
| | Finite Element Analysis Basics | 5 |
| | Automotive Electronics and Control | 5 |
| | Automobile Construction | 5 |
| | Advanced Heat Transfer | 9 |
| | Automotive Power System Principles | 2 |
| Architecture | Structural Engineering | 21 |

