# OpenReview forum: "RBench: Graduate-level Multi-disciplinary Benchmarks for LLM & MLLM Complex Reasoning Evaluation"
_ICML.cc/2025/Conference — ICML 2025 poster_

### Official Review · Reviewer_rJWX · 2025-03-05

**Overall Recommendation:** 3

**Summary:**

This paper proposes a new benchmark called R-Bench, with features of Comprehensiveness, Difficulty, Multimodality and Multilingualism.
The paper also conducted various experiments on current mainstream LLMs and MLLMs using R-Bench.

**Claims And Evidence:**

Yes. All claims are supported by clear and convincing evidence.

**Essential References Not Discussed:**

Essential paper are cited and discussed in the paper.

**Experimental Designs Or Analyses:**

Yes. I've checked the soundness/Validity of experimental designs and analyses.

**Methods And Evaluation Criteria:**

Yes. The evaluation criterias make sense to evaluate the authentic abilities of LLMs/MLLMs.

**Other Comments Or Suggestions:**

For Tab.7 and Section 3.3, which are discussing The effect of CoT,as mentioned in your paper above, do more test on Reasoning Models may be better. (In Tab.7, the only Reasoning Model is o1-mini.)

**Other Strengths And Weaknesses:**

Strengths:
1. This paper is well-structured and easy to follow for readers.
2. This paper is comprehensive and explain the methodology with great details, e.g. Section 2 shows a lot details in describing the process of data collection.
3. The benchmark proposed in this paper is both multi-disciplinary and multi-lingual, incorporating the strengths of existing benchmarks.

Weakness:
1. The paper doesn't fully explore whether R-Bench needs more reasoning abilities than current benchmark. In this paper, authors use thinking tokens/ thinking time to judge, but they haven't taken dataset biases etc into account.
2. This paper focus heavily on the Accuracy in the benchmark. More detailed breakdown of failure modes would be helpful when evaluating LLMs/MLLMs.

**Questions For Authors:**

1.For Section 3.1, Paragraph 2: Is the pairwise comparison approach sufficiently accurate to determine which benchmark (considered in its entirety) demands higher reasoning capabilities?

2.For Section 3.1, Paragraph 3: You mentioned two aspects, on the one hand based on the reasoning tokens and on the other hand using o1 to judge correctly. My question is how these two aspects impact o1 voting in Table 3?(eg. using a weighted sum?)

3.For Section 3.1, Paragraph 3: Reasoning tokens and reasoning time cannot be simply considered to have a linear relationship. (When calling the API service, the GPU memory bandwidth and computing power of the corresponding service will affect the generation speed.) Furthermore, If the length of the R-Bench problem is longer than the baseline problem, the generated reasoning tokens will most likely increase, so the a comparison might not be fair.

4.For Section 3.3: How to compute Consistency and Accuracy in Figure 5&6?

**Relation To Broader Scientific Literature:**

The paper proposed a new benchmark to evaluate LLMs and MLLMs, which is based on prior benchmarks like MMLU and MMMU. R-Bench made improvements on Comprehensiveness and Multilingualism.

**Theoretical Claims:**

There are no theoretical claims in the paper.

---

> ### Author Rebuttal · Authors · 2025-03-30
>
> Thank you for the selfless efforts and constructive comments for improving the quality of this work. The followings are detailed response to your concerns.
>
> ### Q1:
> The paper doesn't fully explore whether R-Bench needs more reasoning abilities than current benchmark.
> ### A1:
> Evaluating the reasoning demands of a dataset on models is inherently challenging, and there is no standard quantitative metric for doing so. In this work, we employed both expert scoring and model-based evaluation to assess the reasoning requirements of our dataset.
>
> Regarding the dataset biases, we understand this as referring to biases across different subjects. To address it, we conducted experiments focusing only on 50 mathematical questions from both R-Bench and MMLU to control for subject-specific variation in reasoning demands. The results showed that the average number of reasoning tokens for R-Bench was 6,867.1, while for MMLU it was 1,011.3, indicating a difference of approximately 6.8 times.
>
> In addition, we asked the OpenAI o1 model to vote on which question required more reasoning. The voting results were 43:5:2 (win:loss:tie) in favor of R-Bench, showing a significant preference for R-Bench questions in terms of reasoning demand.
>
> ### Q2:
> More detailed breakdown of failure modes would be helpful when evaluating LLMs/MLLMs.
>
> ### A2:
> We have conducted analysis of some error examples. Due to format and length limitations during the rebuttal stage, we are unable to include them here. In our analysis of GPT-4o and o1's errors, we observed that most failures occurred during the reasoning process. These errors stem from various sources, such as calculation, flawed reasoning strategies, and perception errors. Notably, the models rarely failed due to a lack of knowledge, indicating that they have generally mastered knowledge at a graduate level. In the revised version, we will include an error analysis section to present our findings.
>
> ### Q3:
> For Table 7, which analyze the effect of CoT, it would be more convincing to include additional reasoning models beyond o1-mini.
>
> ### A3:
> Thank you for your constructive feedback. Due to experimental cost, we only added DeepSeek R1 without CoT. It scored 60.7% on R-Bench, just 0.5% lower than the CoT-enabled version (61.2%). We observed that this performance drop is smaller than that of chat models, where the decrease often exceeds 1% or even 2%.
>
> This is an interesting observation, and we plan to deeply explore CoT's impact across more reasoning and chat models in future work.
>
> ### Q4:
> In Section 3.1, Paragraph 2: Is pairwise comparison sufficient to assess overall benchmark reasoning demand?
>
> ### A4:
> As far as we know,  there is no clearly defined metric for assessing reasoning abilities, whether for humans or foundation models. A single metric, such as the expert pairwise comparison, is one dimension, but it is not sufficiently accurate to determine which benchmark demands higher reasoning capabilities. Our goal is to provide a comprehensive assessment, presented in Tables 2–4, to reflect, across multiple dimensions, that R-Bench imposes higher reasoning demands compared to other benchmarks, both in terms of expert scores and model performance.
>
> ### Q5:
> For Section 3.1, Paragraph 3, how these two aspects impact o1 voting in Table 3?  (eg. using a weighted sum?)
> ### A5:
> We apologize for the confusion caused by Section 3.1, Paragraph 3 in our manuscript. The o1 voting and reasoning tokens are separate and presented in Table 3 and Table 4, respectively. For Table 3, we randomly paired one R-Bench and one MMMU question, then asked o1 which required more reasoning. R-Bench scored 1 point if selected, otherwise MMMU. This was repeated for 30 pairs to compute win rates.
>
> ### Q6:
> Section 3.1, Paragraph 3: Reasoning tokens and reasoning time are not linearly correlated. Additionally, longer R-Bench problems may naturally yield more tokens, making the comparison potentially unfair.
>
> ### A6:
> To strengthen our results, we conducted a reasoning tokens experiment using 50 math questions from both R-Bench and MMLU. OpenAI o1 generated an average of 6,867.1 tokens for R-Bench and 1,011.3 for MMLU—a 6.8× difference. While problem length may play a role, we believe the gap mainly reflects R-Bench’s higher reasoning demands.
>
> ### Q7:
>
> For Section 3.3: How to compute Consistency and Accuracy in Figure 5&6?
>
> ### A7:
>
> Figure 5 shows models' consistency on English and Chinese versions of identical difficult questions. For a question, if both the English and Chinese versions of the question are answered correctly or incorrectly, we increment the consistency variable c = c + 1, Finally, consistency = c / total questions.
>
> For the accuracy in Figure 6, to obtain this accuracy, we grouped the questions by different departments. We then separately calculated GPT-4o's accuracy on the questions from each department to generate Figure 6.

---

> > ### Comment · Reviewer_rJWX · 2025-04-02
> >
> > I have read the authors' rebuttal. They have addressed most of my concerns. However, I am not satisfied with the response to Q6. You acknowledge that problem length may play a role, yet you provide no quantitative comparison of the average lengths of R-Bench and MMLU questions. Without this information, it is unclear whether the observed 6.8× difference in generated tokens is primarily due to reasoning demands or simply a reflection of longer input lengths in R-Bench.
> >
> > $\textbf{Further response to authors' rebuttal}$:
> >
> > Thanks for your detailed response.  Since I could not find the button to reply directly to the authors, I am providing my further response here.
> >
> > I would prefer to see statistics based on the same subject. Currently, the two examples provided are not very representative, and their difference in subject makes comparison quite challenging. I suggest that the authors first categorize the samples by subject, and then randomly select a sufficient number of pairs from R-bench and MMLU with similar lengths in each category. Subsequently, a statistical comparison of their lengths should be performed. This would help demonstrate the results' generalizability and representativeness.
> >
> > $\textbf{Final response to authors' rebuttal}$:
> >
> > Thanks for your prompt and detailed illustrations. They have addressed my concerns. I have raised my score.

---

> > > ### Author Response · Authors · 2025-04-02
> > >
> > > We are glad that the above responses have addressed most of your concerns. Here we hope to address your concern about Q6, which is the impact of question length on reasoning tokens.
> > >
> > > We re-organize the experiment and controlled the length of the questions. We selected 30 questions from R-Bench and MMLU, and their average question lengths were 218.6 characters and 219.7 characters for R-Bench and MMLU respectively.
> > >
> > > Below are examples from R-Bench and MMLU. The first example is from R-Bench and the second one is from MMLU.
> > >
> > >
> > > ```
> > > A semicylindrical glass with a refractive index $n=\sqrt{2}$ is placed in the air. In a plane perpendicular to the axis of the semicylinder, a light ray is incident at a $45^{\circ}$ angle on the flat surface of the semicylinder. What is the range of angles at which the light ray emerges from the semicylinder?
> > > ```
> > >
> > >
> > >
> > >
> > > ```
> > > A solid sphere (I = 0.06 kg·m^2) spins freely around an axis through its center at an angular speed of 20 rad/s. It is desired to bring the sphere to rest by applying a friction force of magnitude 2.0 N to the sphere’s outer surface, a distance of 0.30 m from the sphere’s center. How much time will it take the sphere to come to rest?
> > > ```
> > >
> > >
> > >
> > >
> > > The experimental results show that the average number of reasoning tokens on R-Bench is 6623.2, while the average number of reasoning tokens on MMLU is 933.2, a difference of about 7.1 times. This shows that even if the lengths are similar, the problems in R-Bench still require more reasoning tokens to solve.
> > >
> > > ------------------------
> > > ------------------------
> > >
> > >
> > > ## Further response to reviewers’ comments
> > >
> > > Thank you for your insightful reply. The difference in subjects may introduce bias to a certain extent.
> > >
> > > To avoid this problem, we compared reasoning tokens for R-Bench and MMLU questions of similar question length (average about 240 characters for both benchmarks) in the physics subject. Similarly, we selected 30 questions from each of the physics subjects of R-Bench and MMLU.
> > >
> > > Below are examples from R-Bench and MMLU. The top two belong to R-Bench, and the bottom two are from MMLU.
> > >
> > >
> > > R-Bench:
> > > ```
> > > The line element of the dynamic spherically symmetric Vaidya spacetime is $d s^{2}=-\left[1-\frac{2 M(v)}{r}\right] d v^{2}+2 d v d r+r^{2} d \theta^{2}+r^{2} \sin ^{2} \theta d \varphi^{2}$. $\nu$ is the advanced Eddington coordinate corresponding to time. Find the event horizon expressed in terms of $M, \dot{M}$.
> > > ```
> > >
> > > ```
> > > In a long and straight wire of length $L$, electrons oscillate in phase with angular frequency $\omega$ and small amplitude $a$. Try to calculate the electric field intensity at a far distance $R$ ($R \gg L$) at an angle $\theta$ to the wire.
> > > ```
> > >
> > > MMLU:
> > >
> > > ```
> > > Consider three identical, ideal capacitors. The first capacitor is charged to a voltage and then disconnected from the battery. The other two capacitors, initially uncharged and connected in series, are then connected across the first capacitor. What is the final voltage on the first capacitor?
> > > ```
> > >
> > > ```
> > > In a certain region, the electric field varies with the radius away from origin by the equation Er = –6r^2 + 4r + 3, where r is given in meters and E in N/C. The potential difference between the origin and the point (3, 4) is ?
> > > ```
> > >
> > > The experimental results show that the average number of reasoning tokens on R-Bench is 6157.5, while the average number of reasoning tokens on MMLU is 1021.2, a difference of about 6 times. This suggests that even though the questions with similar length and same subject, the questions in R-Bench still require more reasoning marks to solve.
> > >
> > >
> > > ------------------------
> > > ------------------------
> > >
> > >
> > > ## Thank you for your professional and insightful comments.
> > >
> > > We will incorporate the discussion from the rebuttal stage into the revised version.

---

### Official Review · Reviewer_PuZC · 2025-03-08

**Overall Recommendation:** 4

**Summary:**

This paper proposes R‑Bench, a benchmark designed to evaluate complex reasoning in language and multimodal models. The dataset spans a wide range of subjects and includes more than 1,000 text-based and 665 multimodal questions. The questions are carefully selected and filtered to ensure that they require deep reasoning rather than simple recall, and are provided in both English and Chinese to test cross-linguistic capabilities. Experiments on various LLMs and MLLMs show that even state‑of‑the‑art models achieve only moderate accuracy, with multimodal reasoning posing a greater challenge than text-only tasks. This benchmark not only highlights current limitations across disciplines and modalities but also provides valuable insights and guidance for future improvements in foundation models’ reasoning skills.

**Claims And Evidence:**

In the conclusion, the authors claim that the benchmark achieves competition-level difficulty comparable to contests such as AIME@2024. However, this assertion is not fully substantiated by the evidence presented in the paper. Although the dataset is derived from graduate-level materials, there remains a gap between the difficulty of graduate-level content and that of olympiad-level challenges.The question samples provided in Fig. 3 indicates that the problems do not approach the complexity expected of olympiad-level tasks. Consequently, the claim appears overstated and lacks clear justification.

**Essential References Not Discussed:**

N/A

**Experimental Designs Or Analyses:**

The experiment design is solid.

**Methods And Evaluation Criteria:**

Yes.

**Other Comments Or Suggestions:**

1. In Appendix A.3, there are two "table 8" in line 654.
2. The caption of figure 6 is confusing. I suggest the authors rewrite it.

**Other Strengths And Weaknesses:**

Strengths:
1. The paper is written in a clear and straightforward manner, with all key steps of data collection and benchmark construction thoroughly documented.
2. The experimental evaluation is comprehensive, testing a wide range of models. The observed shortcomings in multimodal reasoning capabilities align with findings from previous studies.
3. The benchmark’s design—spanning a broad knowledge base and requiring deep reasoning—provides a robust testbed for models. Its multilingual and multimodal versions strengthens its utility and relevance.

Weaknesses:
1. The paper does not provide a thorough discussion or comparison of MMMU-pro, a more robust evolution of MMMU. Including MMMU-pro in experiments (e.g., in Tables 2–4) would strengthen the analysis.
2. The benchmark does not report the performance of human experts, leaving an important baseline unexplored.
3. In Section 3.1, only 30 questions are sampled for the win rate comparison, which is a small sample size that limits the reliability of the experimental results.
4. The failure patterns of LLMs and MLLMs are underexplored. More detailed categorization of errors, such as lacking knowledge, wrong  reasoning steps can provide valuable insights into developing more capable models.

**Questions For Authors:**

1. In the model screening process described in Section 2.3, how was the 2000 reasoning token threshold determined? Can you provide details on the distribution of reasoning tokens in the dataset before filtering in Step 4? Can you also estimate the computational cost of prompting o1 with the entire dataset to obtain the reasoing token number?
2. Regarding Section 2.4, can you elaborate on how the answer options were constructed? Specifically, on line 257 you mention that "we manually adjust the options to ensure a sufficient numerical gap between them." What criteria or standards guide these adjustments? Additionally, when including the "all answers are incorrect" option, were there cases where this was intended to be the ground truth answer?
3. How do human experts perform in the benchmark?
4. In the conclusion, you claim that R‑Bench reaches the difficulty level of olympiad-level competitions. However, given that the dataset is derived from undergrad and graduate courses, which are generally more focused on in-depth knowledge rather than competition-level challenge, can you provide further justification or evidence to support this claim?

**Relation To Broader Scientific Literature:**

The development of such a broad and difficult benchmark thus fills a gap in the literature, providing a tool that can more comprehensively evaluate both the retrieval of knowledge and the capacity for complex reasoning.

**Theoretical Claims:**

N/A. No theories involved in this paper.

---

> ### Author Rebuttal · Authors · 2025-03-30
>
> Thank you for your insightful comments, which are crucial to improving the quality of this work. The followings are detailed response to your concerns.
>
> ### Q1:
>
> In the conclusion, authors claim R-Bench achieves competition-level difficulty, but this is not fully supported by the evidence provided.
>
> ### A1:
>
> Indeed, it is difficult to claim that the difficulty of multidisciplinary questions is directly comparable to that of competition problems.
> In our paper, we aimed to illustrate that for current SOTA models such as o1 and DeepSeek R1, their performance on R-Bench is lower than on AIME@2024. For example, o1 and DeepSeek R1 achieved 74.4% and 79.8% accuracy on AIME@2024, while their scores on R-Bench were 69.0% and 61.2%. This observation served as part of the motivation for our claim.
>
> Besides,  in revised version, we will refine our claim and introduce a clearer qualification: specifically, for current advanced reasoning models such as o1, R-Bench achieves competition-level accuracy comparable to contests such as AIME@2024.
>
> ### Q2:
> The paper does not provide a thorough discussion or comparison of MMMU-pro.
> ### A2:
> In revised version, we will include comparisons with MMMU-Pro in Tables 3 and 4.  In rebuttal stage, we conduct o1 voting and o1 thinking time comparison experiments.
>
> We randomly sample 30 questions in MMMU-Pro and use o1 model  to compare them with questions in R-Bench-M.The voting results were 22:7:1 (win:loss:tie) in favor of R-Bench, showing a significant preference for R-Bench questions in terms of reasoning demand. In addition, for questions in R-Bench-M, the OpenAI o1 model required an average of 91.7s  to generate a response, whereas for questions in MMMU-Pro, it required only 28.1s on average. This further suggests that R-Bench-M demands more reasoning capabilities.
>
> Besides, we will expand the discussion of MMMU-Pro in related work.
>
> ### Q3:
> The benchmark does not report the performance of human experts, leaving an important baseline unexplored.
>
> ### A3:
> We agree that human expert performance is an important baseline. However, establishing such a baseline is extremely challenging. The main difficulty lies in recruiting domain experts from different fields to solve thousands of high-difficulty questions, which is practically demanding. Moreover, a single test run is not statistically meaningful; reliable baselines require averaging over 5 runs on thousands of questions.
>
> Therefore, we plan to conduct this baseline on 3–5 selected subjects, such as computer science, to better reflect the gap between models and human experts.
>
> ### Q4:
> In Section 3.1, 30 questions is a small sample size that limits the reliability of the experimental results.
>
> ### A4:
> For the reasoning model scoring, we extended the number of evaluated questions from 30 to 300. The experimental results were consistent with those reported in the paper. In terms of textual reasoning, R-Bench takes approximately 7 times longer than MMLU. For multimodal reasoning, R-Bench requires roughly 4 times the reasoning time compared to MMMU. We will incorporate this result in the revised version to make our experimental results more convincing.
>
> ### Q5:
> The failure patterns of LLMs and MLLMs are underexplored.
>
> ### A5:
> We have conducted an analysis of some error examples; however, due to format and length limitations during the rebuttal stage, we are unable to include them here. In our analysis of GPT-4o and o1's errors, we observed that most failures occurred during the reasoning process. These errors stem from various sources, such as calculation, flawed reasoning strategies, and perception errors (for multimodal models). Notably, the models rarely failed due to a lack of  knowledge, indicating that they have generally mastered knowledge at a graduate level. In the revised version, we will include an error analysis in Section 3.3 and Appendix to present our findings.
>
> ### Q6:
> How was the 2,000 reasoning token threshold set? What's the pre-filter distribution and o1 cost?
> ### A6:
> The 2,000-token threshold was a heuristic decision. We found that 2,000 tokens served as a reasonable threshold to help distinguish "reasoning-oriented" questions from "knowledge-based" ones. In the revised version, we will include a distribution histogram to illustrate the original reasoning token distribution.. The cost of API call for Openai o1 is about 6,000 USD.
>
> ### Q7:
> What criteria guided the manual adjustments to ensure sufficient numerical gaps? Additionally, was “all answers are incorrect” ever used as the correct answer?
> ### A7:
> We believe that the other answer options should differ from the correct answer by at least 10% or 0.5. This is a heuristic guideline to prevent models from being penalized due to minor approximation errors during the reasoning process.
> Regarding the second question, some "all answers are incorrect" options were used as the correct answer.

---

> > ### Comment · Reviewer_PuZC · 2025-04-01
> >
> > Good work. Remember to include all the changes into the next version of the paper. I have revised my score.

---

> > > ### Author Response · Authors · 2025-04-02
> > >
> > > Thank you for your valuable comments! I appreciate your suggestion and will make sure to include all the changes in the next version of the paper.

---

### Official Review · Reviewer_Gu7t · 2025-03-11

**Overall Recommendation:** 4

**Summary:**

This work proposes R-Bench, a graduate-level multidisciplinary, multilingual benchmark for both LLM and MLLM reasoning evaluation, which has coverage similar to MMLU and MMMU while reaching the difficulty of mathematical competitions such as AIME@2024. The authors evaluate multiple closed-source and open-source models on R-Bench and then observe both the progress and limitations of current models in reasoning.

## update after rebuttal

The authors' response address all my concerns. I decide to maintain my score (which is already high) and acknowledge that the contributions of this paper merit publication.

**Claims And Evidence:**

Yes, the claims made in the submission are supported by clear and convincing evidence.

**Essential References Not Discussed:**

No, I believe the related work is thoroughly discussed.

**Experimental Designs Or Analyses:**

Yes. This work conducts extensive experiments to validate the value of their benchmark.

**Methods And Evaluation Criteria:**

Yes, the proposed benchmark in this work makes sense for the reasoning problems in the literature.

**Other Comments Or Suggestions:**

1. Why have the authors placed the related work in Section 4? I believe it would be more appropriate to present the related work in Section 2.
2. In the Introduction section, the authors should express "**Comprehensiveness**" more accurately. For example, economics and chemistry are also subjects that reflect human intelligence, and therefore, we should also evaluate models' performance in these areas.

**Other Strengths And Weaknesses:**

Strengths: This paper is well-written and the motivation is clearly defined. The proposed R-benchmark is highly important and critical for reasoning tasks.

Weaknesses: The related work in Section 4 should be placed earlier in the paper to assist readers in understanding the context. The authors have not thoroughly discussed the superiority of R-benchmark compared to other benchmarks.

**Questions For Authors:**

1. Why should an ideal assessment for complex reasoning emphasize "**difficulty**"? In most cases, difficulty and ease are relative concepts, and it is unclear how to determine whether a problem is difficult. If a benchmark overly emphasizes difficulty, it may overlook the model's performance on simpler problems, whereas in real-world scenarios, simple problems may outnumber difficult ones. I hope the authors can provide a deeper discussion on this matter.
2. In Section 3.1, the authors conduct experiments comparing reasoning abilities with other benchmarks. Although this experiment validates the stronger reasoning ability of R-benchmark, the authors do not provide an explanation for the result. More high-level discussion on the underlying reasons for this outcome is necessary.
3. Section 3.3 shows significant performance variation across disciplines. However, I am unsure whether this experiment is fair. How do the authors ensure that the difficulty of questions in economics and physics is consistent? If the difficulty across disciplines is not uniform, for instance, if economics questions are particularly difficult while physics questions are relatively easier, then this finding would be meaningless.

**Relation To Broader Scientific Literature:**

The benchmark proposed in this paper can provide better support for the research of LLM and MLLM.

**Theoretical Claims:**

This paper mainly focuses on benchmark and its construction process, and therefore does not provide theoretical claims.

---

> ### Author Rebuttal · Authors · 2025-03-30
>
> Thank you for your insightful comments, which are crucial to improving the quality of this work.  In addition, thank you for your positive evaluation of our work, which has been a great source of encouragement for us. The followings are detailed response to your concerns.
>
> ### Q1:
> The related work in Section 4 should be placed earlier in the paper to assist readers in understanding the context.
>
> ### A1:
> In the revised version, we will move the Related Work section to follow the Introduction and add more analysis and discussion to highlight the advantages of R-Bench compared to existing benchmarks.
>
> ### Q2:
>  In the Introduction section, authors should express "Comprehensiveness" more accurately.
>
> ### A2:
> We want to build R-Bench as a comprehensive benchmark, aiming to cover as many subjects as possible that reflect various aspects of human intelligence such as economics and chemistry. We also hope to provide a more precise or definitive characterization of comprehensiveness. However, it is challenging to offer a clear-cut definition of this concept.
>
> In the revised version, we will attempt to describe this property using more concrete language — for example, by stating that R-Bench includes over 100 subjects or by proposing a quantifiable definition of comprehensiveness Thank you again for your suggestion which will help us make the paper more rigorous overall.
>
> ### Q3:
> Why should an ideal assessment for complex reasoning emphasize "difficulty"? In most cases, difficulty and ease are relative concepts, and it is unclear how to determine whether a problem is difficult.
>
> ### A3:
> Indeed, in real-world scenarios, simple problems may outnumber difficult ones. However, in our context, difficulty refers to the level of challenge a model faces when solving a problem.
>
> Just as humans encounter exams of varying difficulty at different educational stages—elementary school, middle school, university—these assessments guide individual development by signaling what knowledge to acquire and which direction to pursue. Similarly, for foundation models, evaluation benchmarks serve as developmental milestones.
>
> For instance, prior to December 2022 (before the release of ChatGPT), the focus was on whether large models could memorize and reproduce a broad range of knowledge. At that time, MMLU provided a meaningful developmental direction and was considered challenging for models. As model capabilities improved with the release of GPT-4o, Claude 3.5, and others, MMLU became increasingly saturated, i.e., it started to be perceived as easy, prompting the development of more difficult benchmarks like MMLU-Pro.
>
> Now, with the advent of advanced reasoning models such as o1, even MMLU-Pro is showing signs of saturation. This creates a demand for more difficult benchmarks to continue guiding model development.
>
> Based on the above, we define difficulty in terms of the saturation level of current state-of-the-art models. This is exactly what our "o1 saturation" metric in Table 1 aims to capture — a quantifiable measure of how much room there is for improvement for top-performing models.
>
> In this sense, R-Bench not only points to the future direction of model development — emphasizing complex reasoning — but also presents immediate challenges for today’s best models, none of which have yet saturated R-Bench.
>
> ### Q4:
> In Section 3.1, the authors do not provide an explanation for the result. More high-level discussion on the underlying reasons for this outcome is necessary.
>
> ### A4:
> Thank you for pointing out this issue. In the revised version, we will provide more details about Section 3.1 experiments, including the specific implementation, the voting template used during the user study, and additional explanation and analysis of the results. We believe these additions will help improve the quality of our work and make the methodology clearer and more accessible to readers.
>
> ### Q5:
> Section 3.3 shows significant performance variation across disciplines. However, I am unsure whether this experiment is fair. How do the authors ensure that the difficulty of questions in economics and physics is consistent?
>
> ### A5:
> Thank you for your insightful comment. Indeed, it is challenging to ensure that the difficulty level is consistent across different subjects. In the next version, we will make the description in Section 3.3 more rigorous—for example, by revising the subsection title to emphasize that all subjects still require improvement and none have reached a perfect state. Additionally, we plan to include an analysis of error cases across different subjects under this section, to help readers better understand the types of mistakes models make in each subject.

---

### Official Review · Reviewer_KtMe · 2025-03-11

**Overall Recommendation:** 3

**Summary:**

This paper introduces R-Bench, a new benchmark designed to evaluate complex reasoning capabilities in both LLMs and MLLMs. The benchmark contains questions in two languages: English and Chinese. There are 1,094 questions in 108 subjects
for textual evaluation and 665 questions in 83 subjects for multimodal evaluation. The authors conduct comprehensive experiments of current LLMs and MLLMs.  Several key observations are presented for further development.

**Claims And Evidence:**

Yes. The claims made in the submission are supported by clear and convincing evidence.

**Essential References Not Discussed:**

No

**Experimental Designs Or Analyses:**

Weaknesses:
1. The error analysis part provides very few valuable insights. The observations in Section 3.3 could deliver more fine-grained observations into the reasoning process of GPT-4o or o1. How these models make mistakes is a good topic to discuss in the error analysis section. For example, does the model obtain a wrong answer due to a deduction error, knowledge error, or other error types?

**Methods And Evaluation Criteria:**

Strengths of the benchmark:
1. The benchmark dataset is of high quality. The authors make great efforts to ensure the difficulty of the dataset.
2. The benchmark contains data newly collected from college courses.
Weaknesses of the benchmark and its evaluation method:
1. The scope of the benchmark is very comprehensive, with 108 subjects in the dataset. However, compared with the coverage of the subjects, the total number of the collected questions seems rather limited, with only 1094 and 665 questions for textual and multimodal questions.  For example, in Table 8, there are many sub-subjects with questions less than 5. This could lead to randomness when evaluating.
2. The evaluation strategy seems to be too straightforward. As the dataset targets difficult reasoning questions, the authors are expected to provide an evaluation strategy for the reasoning rather than only the answer. Since the problems are all sourced from college courses, the knowledge required when reasoning could also be difficult. Therefore, the model could obtain a wrong answer due to a lack of specific knowledge required in reasoning instead of a poor reasoning capability. Simply assessing the final answer omits this situation totally.

**Other Comments Or Suggestions:**

1. More data samples of different subjects could be placed in the appendix to give the readers a better view of the dataset.
2. Table 1 could add another column demonstrating the dataset size.

**Other Strengths And Weaknesses:**

Weakness:
1. The dataset size seems limited compared with MMLU and MMMU. For example, MMMU contains 11.5K samples for multimodal questions while this dataset contains only over 600 samples.

**Questions For Authors:**

1. I think the authors should clarify how to better evaluate the model prediction. For example, how can errors caused by knowledge be considered and how can the evaluation contain more aspects?
2. Do authors have any plans to expand the dataset size?

**Relation To Broader Scientific Literature:**

The related models and benchmarks have been fully discussed in the related work.

**Theoretical Claims:**

No theoretical claims are introduced.

---

> ### Author Rebuttal · Authors · 2025-03-30
>
> Thank you for your insightful comments, which are crucial to improving the quality of this work. The followings are detailed response to your concerns.
>
> ### Q1:
> Despite the broad subject coverage, the total number of collected questions, 1,094 for textual and 665 for multimodal, is relatively limited.
>
> ### A1:
> We were indeed aware of the issue regarding the number of test questions when designing R-Bench. We would like to address this concern from following three aspects:
>
> 1. In fact, we originally collected over 15,000 candidate questions—comparable to benchmarks like MMLU and MMMU. However, our rigorous and multi-stage filtering process eliminated a large portion of these samples to ensure high quality, fairness and difficulty. This strict curation results in a smaller final dataset, but we believe it contributes to a more reliable evaluation. In future versions of R-Bench, we plan to include more questions that require advanced reasoning and offer broader topic coverage to further improve the benchmark.
>
> 2. If we consider a typical human examination paper to contain around 20 questions, then R-Bench's 1,094 questions would be equivalent to approximately 55 such exams, while the 665-question subset corresponds to about 33 exams. Compared to human-oriented standardized tests such as the ACT (American College Test), SAT (Scholastic Assessment Test), GRE (Graduate Record Examination), and China's GAOKAO, R-Bench significantly increases the number of test questions and the overall coverage.
>
> 3. Based on the evaluation results, although R-Bench contains only 1,094 and 665 samples for language and multimodal model testing respectively, the outcomes align well with our understanding of the reasoning capabilities of different models. For example, the OpenAI o1, o1-preview and o1-mini models demonstrate stronger performance than GPT-4o and DeepSeek R1 exhibits stronger reasoning ability than DeepSeek V3.
>
> In addition, the results also indicate that most models still require significant improvements in handling complex reasoning problems. This provides new insights and guidance for the future development of reasoning-capable models.
>
> ### Q2:
> The evaluation strategy seems to be too straightforward. As the dataset targets difficult reasoning questions, the authors are expected to provide an evaluation strategy for the reasoning rather than only the answer.
>
>
> ### A2:
> During the development of R-Bench, we also considered incorporating evaluation methods beyond final outcome, such as process-based evaluation. However, due to the inherent uncertainty in reasoning processes, it was difficult to identify a reliable way to assess intermediate steps consistently. As a result, we chose to reflect the focus on reasoning primarily through our question selection process. We applied rigorous filtering involving both domain experts and intelligent models to ensure that the majority of questions emphasize reasoning over knowledge, rather than being heavily knowledge-dependent. In future work, we plan to explore alternative evaluation methodologies as well. We believe your suggestion provides a valuable direction for improving our benchmark.
>
>
> ### Q3:
> The error analysis part provides very few valuable insights. The observations in Section 3.3 could deliver more fine-grained observations into the reasoning process of GPT-4o or o1.
>
> ### A3:
> We agree that it can help improve the quality of our work. We have conducted an analysis of some error examples; however, due to format and length limitations during the rebuttal stage, we are unable to include them here. In our analysis of GPT-4o and o1's errors, we observed that most failures occurred during the reasoning process. These errors stem from various sources, such as calculation mistakes, flawed reasoning strategies, and perception errors (in the case of multimodal models). Notably, the models rarely failed due to a lack of factual knowledge, indicating that they have generally mastered knowledge at a graduate level. In the revision, we will include an error analysis section in Section 3.3 and the supplementary materials to present our findings and insights. We believe this will strengthen our work, and we sincerely appreciate your professional and helpful feedback.
>
> ### Q4:
> Do authors have any plans to expand the dataset size?
> ### A4:
> Yes, we do have plans to expand the size of the dataset. Our expansion will not be limited to multi-disciplinary reasoning problems; we also plan to extend towards more general-purpose reasoning tasks — for example, complex reasoning scenarios in daily life such as path planning.In future work, we hope to scale the dataset to around 5,000 high-quality questions in future work.
>
> ### Q5:
> For other comments or suggestions
>
> ### A5:
> Due to the length limit of the rebuttal, we apologize for not being able to give a detailed response. We will follow your two insightful suggestions in the revision to improve the quality of our work.

---

> > ### Comment · Reviewer_KtMe · 2025-04-02
> >
> > I have read all the authors' rebuttal. However, my concern about evaluation still exists: More fine-grained evaluation is important, especially for a benchmark targeting complex reasoning evaluation, as the title says. Only evaluating the correctness of the final answer provides very little insight into the models' reasoning behavior.
> >
> > Overall, I think the introduction of the benchmark is meaningful, but the evaluation method and provided insights are somewhat limited. I will maintain my score.

---

> > > ### Author Response · Authors · 2025-04-02
> > >
> > > Thank you for your valuable suggestions and positive score for our work. We highly appreciate the improvement directions you have suggested. We will continuously improve this work according to your comments in the future.

---

### Decision · Program_Chairs · 2025-05-01

**Decision:**

Accept (poster)

**Comment:**

This work proposes R-Bench, which is a reasoning benchmark targeting graduate-level tasks across multiple disciplines and languages. The benchmark is designed for both LLM and MLLM evaluation. It achieves coverage similar to MMLU and MMMU, while introducing more challenging problems inspired by math competitions. The authors test a series of open-/closed-source models, which provides key insights into current reasoning performance and limitations.

During the initial review phase, all four reviewers acknowledged that the paper offers meaningful insights to the foundation model research community. At the same time, they raised several concerns, regarding the scale and construction of the dataset, as well as comparisons with more recent baselines. Following the rebuttal, all reviewers submitted acknowledgments and reached a consensus to recommend acceptance.

AC has carefully reviewed the paper and the reviewers’ comments, and agrees with the overall recommendation. The authors are encouraged to incorporate the reviewers’ constructive feedback in the final version to further strengthen the clarity and impact of this work.